# How Effective Are Antimicrobial Agents on Preventing the Adhesion of *Candida albicans* to Denture Base Acrylic Resin Materials? A Systematic Review

**DOI:** 10.3390/polym14050908

**Published:** 2022-02-24

**Authors:** Salwa Omar Bajunaid

**Affiliations:** Department of Prosthetic Dental Science, College of Dentistry, King Saud University, Riyadh 4545, Saudi Arabia; sbajunaid@ksu.edu.sa; Tel.: +966-5900-28784

**Keywords:** antimicrobial agents, candida biofilm, denture stomatitis, denture acrylic, surface properties

## Abstract

Denture stomatitis is a common oral infection caused by Candid albicans. It occurs under removable dentures due to several causative and contributing factors. If not treated, it can lead to fatal systemic candida infections. Different materials and techniques have been used to treat denture stomatitis; however, no single treatment has succeeded. The purpose of this study was to review novel techniques for incorporating antimicrobial and protein repellent agents into denture acrylic resin materials and their effect on the adhesion of *Candida albicans* to denture base acrylic resin materials to prevent and/or treat denture stomatitis. A systematic review was conducted in which an electronic search was undertaken using different databases and relevant keywords. The literature search revealed numerous studies describing different antifungal materials incorporated into different denture acrylic resin materials. The investigated materials showed significant antimicrobial activity with slight or no effect on the physical and mechanical properties; however, the optical properties were particularly affected with higher concentrations. The incorporation of antimicrobial agents to reduce or prevent *Candida albicans* biofilm formation on acrylic resin proved to be very effective, and this effect was found to be proportional to the percentage of the material used. The latter should be considered carefully not to alter the physical, mechanical or optical characteristics of the denture base material.

## 1. Introduction

The oral cavity harbors microbiota which consist of thousands of different microorganisms essential to protect against infectious agents that can attack the human body [1]. However, certain conditions including immunosuppression, malnutrition, poor oral hygiene, misuse of antibiotics, trauma and the misuse of removable prosthesis, may increase the risk of developing oral infections [2]. For example, under normal circumstances, *Candida albicans* is a harmless microorganism that constitutes part of the normal flora of the skin as well as the mucous membranes of the mouth, ear, nose, eyes, reproductive tract and other organs of the body. However, immunosuppression can stimulate the overgrowth of these organisms and the occurrence of opportunistic infections known as candidiasis or candidosis localized in the oral cavity [3].

Denture stomatitis is a very common example of oral candidiasis and is manifested as inflammation of the mucosa underlying a complete or a partial removable prosthesis [4]. It is more commonly found on the palatal mucosa of complete denture wearers (particularly older females), medically compromised and immunocompromised individuals, and is often associated with angular cheilitis. Studies showed that denture stomatitis affects between 11–67% of complete denture wearers [5].

Fungal microbes, particularly *C albicans*, are attributed as the most common cause of denture stomatitis. The Candida biofilm adheres to the fitting surface of the denture and proliferates with poor oral hygiene [4]. Different classifications have been proposed for denture stomatitis, but the reference classification is the one suggested by Newton in 1962 [5]. It is based exclusively on classical clinical signs; Newton’s type I is characterized by localized simple inflammation or pinpoint hyperemic lesions; type II shows diffuse erythema confined to the mucosa contacting the denture, and Newton’s type III is known to show papillary inflammatory hyperplasia of the keratinized mucosa.

Local factors contributing to the development of denture stomatitis include decreased salivary flow/xerostomia, ill-fitting removable prosthesis, allergic reaction and irritation to the denture base material and the presence of candida in the mucosa [6]. Properties of the denture surface such as surface roughness (Ra) and hydrophobicity play important roles in the adhesion of plaque, microbial biofilms and hence denture stomatitis. An Ra level less than 0.2 μm is desirable to prevent the microbial adhesion to the denture surface as rough denture surfaces entrap the microorganisms and make their removal more difficult during denture cleaning. Furthermore, the hydrophobic interaction between *Candida albicans* and the denture base increases the adherence of the candida biofilm [7].

Systemic predisposing factors such as dietary deficiency and hematological disorders also play a role, although less direct, in the development of denture stomatitis. In most cases, therefore, treatment of denture stomatitis should focus first on the elimination of the denture faults, discontinue denture wearing, sanitization of the existing denture and/or fabrication of a new denture [3]. According to Greenspan et al., if left untreated, denture stomatitis may lead to systemic infection, especially in immunocompromised patients where relapse is common [8].

Different treatment approaches have been proposed at both the local and systemic levels. These include the use of local and systemic antifungal agents (their efficiency is impaired by the emergence of drug-resistant *Candida* species), correcting the ill-fitting denture, soaking the denture in antifungal disinfectants, using laser to treat the affected mucosa, or a combination of these approaches in some cases [8]. Emami et al. conducted a systematic review and compared the different available treatment approaches of denture stomatitis. They found no obvious differences between disinfection methods of the denture and the local or systemic use of antifungal agents [9]. The high prevalence of denture stomatitis highlights the need for developing new therapeutic strategies such as adding or incorporating different antifungal agents into the denture acrylic resin materials [4]. Although numerous treatment methodologies have been proposed to treat denture stomatitis, a gold standard treatment has not yet been identified.

This review aims to focus on presenting novel techniques of incorporating different antimicrobial biomaterials and protein repellent agents to removable denture acrylic resin base materials to inhibit *Candida albicans* biofilm formation and growth on the denture surface and prevent and/or treat denture stomatitis.

## 2. Materials and Methods

A systematic review was conducted in which a structured electronic search of bibliographic indexed databases including PubMed Central, Google Scholar, Elsevier’s ScienceDirect and the Cochrane Database was conducted using the following key words: “adhesion”, “antifungal agents”, “biofilm formation”, “biomaterials integration”, “candida-albicans, “candidiasis”, “dental prosthesis”, “denture stomatitis”, “PMMA”, “polymethyl methacrylate”, “denture acrylic resin”, “saliva” and “surface properties.” Inclusion and exclusion criteria are summarized in Table 1. The search was supplemented with manual searching and cross referencing of retrieved literature. Abundance of papers were found; then titles and abstracts were further read to extract relative literature based on the inclusion criteria.

## 3. Results

The electronic search retrieved an abundance of papers presenting different traditional and contemporary medications and techniques to treat denture stomatitis. Titles and abstracts were screened by the author for possible inclusion in the review. Articles that did not meet the inclusion criteria were excluded, and relevant articles were further read; only those relevant to the addition and incorporation of antifungal or protein repellent agents into denture acrylic resin materials were included in this review.

Several published reviews indicated that different antimicrobial polymers with inherent antifungal properties can be coated on or incorporated within denture acrylic resin, denture liners or denture adhesives. Nanocomposites are formed by adding nanofillers or nanoparticles with antibacterial and antifungal effects to the denture acrylic resin base material. The resultant modified denture base material aims to resist microbial biofilm formation and growth and thereby prevent denture stomatitis [10]. Although the mentioned materials or techniques showed significant cytotoxicity to the oral microbes including *Candida albicans*, higher concentrations of some of these incorporated nanomaterials altered some of the physical, mechanical and optical properties of the denture base material.

For the ease of readability of this review, the author divided these antimicrobial agents into different categories as follows:

### 3.1. Polymers

Based on their nature, these polymers were categorized into three categories: (i) polymeric biocides, (ii) biocide-releasing polymers, and (iii) biocidal surface coatings as shown in Table 2 [11].

#### 3.1.1. Polymeric Biocides

Examples of Polymeric Biocides are methallyl phosphate monomers, methacrylic acid monomers. Several investigations were conducted on these biocides and showed significant reduction of the adhesion of *Candida albicans* to the acrylic denture surface with no effect on the physico-mechanical and optical properties of the resin [12,13,14]. This reduction is attributed to the negative charge formation on the denture base which produces a repulsive force against the negatively charged surfaces of the bacteria and reduces its adherence. Rodriguez LS et al. tested the addition of 2-tert-butylaminoethyl methacrylate (TBAEMA), an example of methacrylic acid monomers; they found significant antimicrobial activity against *S. aureus* and *S. mutans* biofilms, but no significant effect on *Candida albicans.* The incorporation of TBAEMA into the polymethyl methacrylate acrylic monomer (PMMA) enables the negatively charged amino groups to arise on the surface of the acrylic resin and results in disorganization of the cell membrane of the bacteria and cell lysis. However, this modification demonstrated altered physical and optical properties by increasing surface roughness and reducing the flexural strength of the resin, particularly when the concentration of the polymer is higher than 1.75% [15].

#### 3.1.2. Biocide-Releasing Polymers

Biocide-releasing polymers are nanosized metal oxides that exhibit fungistatic and/or fungicidal activities such as silver oxide, silver zeolites, zinc oxide, zirconium oxide, titanium dioxide, and others (Table 2). Gad M et al. and Siedenbiedel F et al. reviewed the effect of adding these metal oxides to polymethyl methacrylate (PMMA) acrylic resin and demonstrated excellent prevention of bacterial and fungal biofilm colonization, and potential prevention and treatment of oral candidiasis [16,17]. The addition of these metal fillers was found to improve the physical and mechanical properties of the PMMA material, as well as its thermal conductivity and thermal stability [18,19,20].

##### Silver Nanoparticles

Silver nanoparticles showed significant antibacterial activity (proportional to the percentage of the added nanoparticles) by inhibiting the replication of the microorganism’s cell. It functions as a carrier for biocides and releases them close to the surface of the cell [21]. Its addition to the acrylic resin enhanced the physical and mechanical properties of PMMA and improved its thermal conductivity and compressive strength [22,23]. However, another study showed that the incorporation of silver nanoparticles resulted in increased surface hardness, flexural modulus and a slight decrease of the flexural and impact strength [24]. Despite their significant antifungal activity, several studies reported that their incorporation into PMMA acrylic resin caused resin discoloration and interfered with the esthetics of the removable prosthesis [2,25]. The AgNPs-modified acrylic resin showed higher antifungal activity against C. glabrata and S. mutans than against *Candida albicans* [26].

##### Silver Zeolites

Silver zeolites are aluminum silicate crystals with interconnected pores that can host cations (such as Ag) and exchange them with other cations from the environment. When the free cations contact the microorganisms, they inactivate the microbial enzymes and inhibit cell replication [27]. Silver zeolites (SZ) were incorporated into dental resins, glass ionomer cements and tissue conditioners and proved to provide antibacterial activity. Casemiro et al. investigated the antimicrobial activity of PMMA denture acrylic resin incorporated with different concentrations of SZ and found that the modified resin showed good antimicrobial activity; however, higher concentrations of the SZ negatively affected the mechanical properties of the resin [28].

##### Zinc Oxide

Kamonkhantikul et al. and Anwander et al. evaluated the effect of incorporating ZnONPs into heat-cured denture base resins on antifungal, mechanical and optical properties of the resin. The addition of zinc oxide with or without methacryloxypropyltrimethoxysilane (salinization) was tested and the results showed that the salinized group had better reduction of *Candida albicans* adherence, less color change (ΔE) and no change of mechanical properties to the denture base relative to the non-salinized group [10,29].

##### Zirconium Oxide

Hamid et al. investigated the effect of adding ZrO_2_NPs to PMMA denture base acrylic resin to inhibit and/or treat denture stomatitis and they found that this incorporation had a significant long-term antifungal effect [30]. In contrast, another study by Abualsaud et al. found insignificant effect of ZrO_2_NPs on the reduction of *Candida albicans* biofilm formation [31]. While the incorporation of ZrO_2_NPs improved the tensile strength significantly (proportional to the concentration of the nanoparticles), it adversely affected the surface roughness and the translucency of the acrylic resin [32]. Zirconia nanoparticles have been also added to cold-cured acrylic resin (used for denture repair) and their influence on *Candida albicans* biofilm adhesion was tested. The results indicated that this addition of ZrO_2_NPs significantly reduced the adhesion of the biofilm to repaired and cold-cured acrylic resin and had a significant antifungal effect [33].

#### 3.1.3. Biocidal Surface Coating (Protein Repellent Agent)

Bajunaid et al. investigated the influence of incorporating protein repellent agent (2-methacryloyloxyethyl phosphorylcholine-MPC) into a high-impact denture acrylic resin. They studied the effect of this modified acrylic resin on candida biofilm formation and on the surface roughness of the acrylic resin. The researchers found that adding MPC at 4.5% wt resulted in a significant fungal retardation and no effect on the surface roughness of the developed acrylic resin. The authors suggest that the newly developed denture material has the potential to prevent denture stomatitis (Figure 1 and Figure 2) and Table 2 [34]. Another study investigated the effect of the dual incorporation of MPC as a protein repellent agent and dimethylaminohexadecyl methacrylate (DMAHDM) as an antifungal agent to prevent the adherence of *Candida albicans* to the denture base material. This dual incorporation of MPC and DMAHDM reduced *Candida albicans* biofilm colony-forming unit by two orders of magnitude. Although the addition of MPC and DMAHDM alone or in combination significantly reduced the flexural strength of the material, they showed reduced roughness values when compared to control groups (Figure 3) [35].

### 3.2. Nano-Diamonds and Fillers

#### 3.2.1. Nano-Diamonds

Are nanocarbon agents that are superior to other metal/metal-oxide nanoparticles. They are biocompatible, more chemically stable, have excellent physical and chemical properties, antibacterial activities and do not induce cytotoxicity. They are used in dentistry, such as in dental implant coatings, guided tissue regeneration and polymer reinforcement [36,37]. The influence of the incorporation of nanodiamonds (NDs) to polymethyl methacrylate (PMMA) on fungal adhesion and mechanical properties of the resin was tested by several authors. This modification of the resin resulted in improved flexural strength, elastic modulus, surface roughness and surface hardness of the resin and showed significant reduction of fungal adhesion and viability. However, the impact strength of the resin was decreased (Table 2) [38,39,40].

#### 3.2.2. Fluoridated Glass Fillers

These fillers act as a fluoride reservoir and release fluoride ions. Surface reaction-type pre-reacted glass-ionomer (S-PRG) exhibits fungistatic effects against *Candida albicans* and can release six types of ions (Na^+^, Sr^2+^, SiO_3_^2−^, Al^3+^, BO_3_^3−^ and F^−^) [41]. Researchers evaluated the effect of its incorporation into PMMA denture base resin on candida growth and surface roughness of the resin and found significant reduction of the biofilm formation and prevention and treatment of denture stomatitis. A slight increase in the surface roughness of the acrylic resin was also detected (Table 2) [42].

### 3.3. Natural Antifungal Agents

Adding natural extracts as fillers in the denture base resin was evaluated in the literature and found to be effective in preventing microbial adhesion and proliferation. Despite the excellent antifungal properties of these natural extracts, they showed adverse effects on the physical and mechanical properties of the modified acrylic resin, Table 2 [43].

#### 3.3.1. Chitosan

Chitosan is a biocompatible, biodegradable, antioxidant, antibacterial, antifungal and antitumor polymer obtained from the hard outer skeleton of shellfish and is used in medicine and dentistry. It has been incorporated into mouthwashes, denture cleansers and denture resin material [44]. Researchers evaluated the antifungal ability of chitosan quaternary ammonium salt added to denture acrylic resin and its effect on the tensile strength of the acrylic resin. Chitosan showed good antimicrobial ability and had no effect on the tensile strength of the resin. More than 50% reduction of biofilm growth was noticed after 3 h and greater inhibition was observed at 18 h of incubation [45,46]. Moreover, the effect of adding different concentrations of chitosan to denture acrylic resin on the physical properties of the resin including flexural strength (FS), fracture toughness (FT), impact strength (IS) and surface roughness (Ra) was investigated. The results indicated that the chitosan incorporation improved the flexural strength, fracture toughness and impact strength when added at 5% wt and the surface roughness when added at 15% wt. However, the surface roughness increased when the chitosan was added at 5% and 10% wt [47].

An in vivo study measured the salivary pH, uric acid and C-Reactive protein levels in 15 patients wearing a complete denture containing chitosan nanoparticles. Results were compared to control dentures that did not have the chitosan nanoparticles. Results confirmed other in-vitro studies which showed that chitosan nanoparticles have an antimicrobial effect against *Candida albicans* and can thus prevent denture stomatitis [48].

#### 3.3.2. Henna (*Lawsonia inermis*)

Henna is an inexpensive natural extract that has antifungal properties. It is safe and has no unwanted health side effects other than potential allergic reactions. It has been reported to have analgesic, hypoglycemic, anti-inflammatory, antibacterial, antimicrobial, antifungal, antiviral and antiparasitic properties [49,50]. Nawasrah et al. found that adding henna to acrylic denture base material may control the proliferation of *Candida albicans* and hence help to inhibit or treat denture stomatitis [51]. However, Gad and colleagues evaluated the flexural strength of acrylic resin denture base material modified with henna and found that the flexural strength was negatively affected and decreased significantly. This decrease was proportional to the concentration of henna added [52].

#### 3.3.3. Neem Powder (*Azadirachta indica*)

Neem powder is another natural product that has antimicrobial properties. Its incorporation into acrylic resin resulted in the formation of a composite that significantly decreased the Candida albicans adhesion and the candida count was conversely proportional to the concentration of the neem powder [53]. The effect of leaching of neem powder and fluconazole added to auto-polymerized, heat-cured acrylic resins and silicone soft liner on *Candida albicans* formation and growth was tested. It was found that fluconazole had greater antifungal activity against *Candida albicans* than the neem powder. The soft liner showed the longest sustained leach of the antifungal agents [54].

### 3.4. Antifungal Medicaments

The effect of impregnation of fluconazole (FLU) or chlorhexidine (CHX) into autopolymerizing or heat-cured methacrylate based acrylic resin on treatment of denture-induced oral candidiasis was investigated. The results conferred that fluconazole had a poor inhibitory effect while chlorhexidine had high antifungal activity and released steadily out of the autopolymerized acrylic resin. The amount leached out demonstrated antifungal activity against *Candida albicans* [55,56].

Maluf et al. found that polymethyl methacrylate (PMMA) acrylic resins modified by the incorporation of chlorhexidine diacetate inhibited the growth of *Candida albicans* biofilms. The authors further evaluated the antifungal effect of this acrylic resin modified with chlorhexidine in the oral cavity of 32 participants and results indicated sustained release of chlorohexidine and reduction of the adhesion of the microbial biofilm formation, Table 2 [57,58].

**Table 2 polymers-14-00908-t002:** Different antifungal and protein repellent agents incorporated into PMMA denture acrylic resin materials.

Type of Material			Author	Date
Polymer	Polymeric Biocides		Pesci-Bardon et al.	2006
			Raj, P.A. and Dentino	2011
			Rodriguez LS et al.	2013
			da Silva Barboza et al.	2021
	Biocide-releasing polymers	Silver nanoparticles	Sehajpal	1989
			*Siedenbiedel F* and Tiller	2012
			Yadav et al.	2012
			Asar et al.	2013
			Gad M et al. 2017	2017
			Hamedi-Rad et al.	2014
			Ghafari et al.	2014
			Suganya et al.	2014
			de Castro et al.	2016
			Zhang et al.	2017
			Hashim et al.	2020
		Silver zeolites	Kawahara	2000
			Casemiro et al.	2008
		Zinc oxide	Kamonkhantikul et al.	2017
			Anwander	2017
		Zirconium oxide	Gad et al.	2017
			Gad et al.	2018
			Hamid et al.	2021
			Abualsaud et al.	2021
	Biocidal surface coatings (Protein Repellent Agent)		Bajunaid et al.	2021
			Bajunaid et al.	2022
Nano-diamonds and Fillers	Nano-diamonds		Mangal et al.	2019
			Al-Harbi et al.	2019
			Fouda et al.	2019
	Fluoridated glass fillers		Kamijo et al.	2009
			Tsutsumi et al.	2016
Natural Antifungal Agents	Chitosan		Song et al.	2015
			Ikono et al.	2019
			Fakhri et al.	2020
			Sonawane and Kamb	2020
			Chander and Venkatraman	2021
	Henna (*Lawsonia inermis*)		Nawasrah et al.	2016
			Gad et al.	2018
	Neem powder (*Azadirachta indica*)		Hamid et al.	2019
			Chincholikar et al.	2019
Antifungal Medicaments			Darwish and Amin	2011
			Salim et al.	2013
			Maluf et al.	2019
			Maluf et al.	2020

## 4. Discussion

The prevalence of fungal infections has increased progressively among the aging immunocompromised population. Causes include resistance to antimicrobial drugs and restrictions on the use of these drugs due to their unwanted side effects [59]. Denture stomatitis is an example of oral fungal infections caused by *Candida albicans.* Different antifungal materials and agents have been used and the high incidence of denture induced stomatitis encouraged the search of new antifungal materials and agents [4,60].

The literature showed the abundance of papers and experiments of many different antifungal nanoscale polymers that were added to heat-cured, cold-cured acrylic resin, denture liners and denture adhesives. This review identified the novel means of incorporating antimicrobial and protein repelling agents within removable denture acrylic base materials to develop a modified denture base with antifungal properties to reduce or inhibit the formation and growth of *Candida albicans* biofilm on the denture surface.

These antifungal agents include nanoparticles, monomers, metal oxides, glass fibers, chlorhexidine, protein repelling agents or naturally occurring sugars and herbal extracts such as chitosan, henna or neem powder. All these materials and their derivatives proved to be highly significant in inhibiting and reducing *Candida albicans* accumulation and growth. However, there were noted negative effects on the mechanical, physical and optical properties of the denture acrylic resin. These negative impacts were found to be proportional to the concentration of the antimicrobial agent.

Heat-cured acrylic resins are commonly used to fabricate partial and complete removable dentures due to their low cost and good esthetics [60]. This desirable esthetic of the prostheses should not be compromised by the addition of antimicrobial agents. When coated, the denture surface might lose its optical properties overtime due to the scratches developed on the surface by the mechanical cleansing of the denture. These scratches will harbor microbial adhesion and cause staining. The incorporations of the antifungal agents range in their effects on the optical characteristics of the acrylic denture base. While monomers and glass fillers show very little if any color changes, the TiO_2_ nanofillers cause white discoloration and the AgNPs and natural extracts result in a gray discolored denture [23,51,61,62]. These materials can be applied to less visible areas of the denture base such as posterior and lower lingual areas.

Although the addition of antimicrobial agents to denture base materials proved to be significantly effective in developing antimicrobial denture material, the concentration of these materials should be studied so as not to jeopardize the biocompatibility, physical or optical characteristics of the denture resin base. Further experimental and in-vivo studies to simulate oral conditions are needed to test the influence of incorporation of different antifungal materials into different denture base materials, denture reline materials and denture adhesives.

## 5. Conclusions

This review summarized different techniques of incorporating different antimicrobial and protein repellent agents to inhibit the adhesion of fungal species (particularly *Candida albicans*) on denture acrylic resin base material to prevent denture stomatitis. This review addressed these materials and the effect of their incorporation on the physical, mechanical and optical properties of the acrylic resin material. The author concluded that those material had great and very efficient antifungal effect when added to denture acrylic resin materials with minimal or no effect on the physical, mechanical or optical properties of the developed acrylic resin.

## Figures and Tables

**Figure 1 polymers-14-00908-f001:**
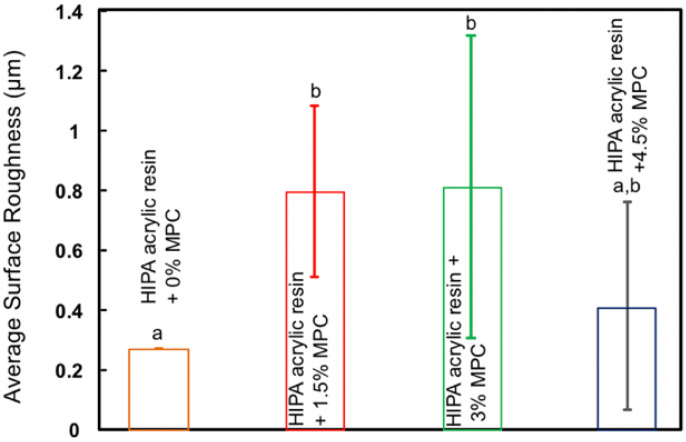
Average roughness values of HIPA acrylic mixed with 0% MPC, 1.5% MPC, 3% MPC and 4.5% MPC as a protein repellent (mean ± SD; *n* = 12). HIPA acrylic resin with 4.5% MPC, showed surface roughness values similar to that of the control group (*p* > 0.05).

**Figure 2 polymers-14-00908-f002:**
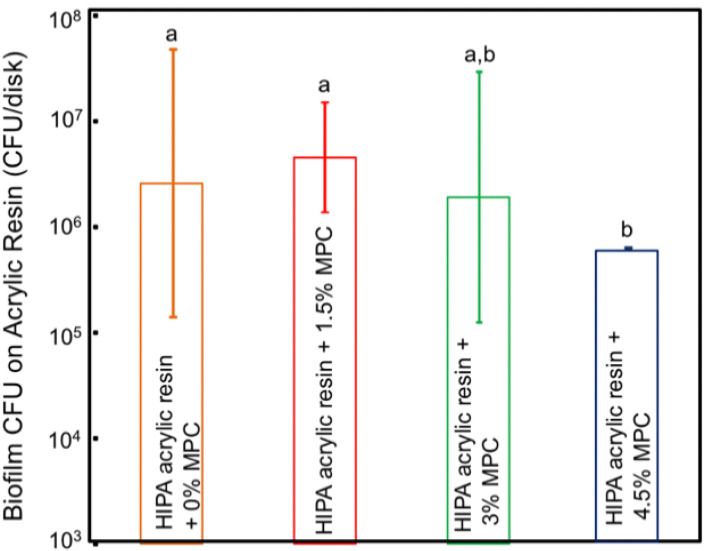
Colony-forming unit (CFU) counts of the *C.albicans* 2-day biofilm on the HIPA acrylic disks (mean ± SD; *n* = 4). Experimental group with 4.5% MPC resulted in a significant ≅ 1 log CFU reduction, compared to control group with 0% MPC (*p* < 0.05).

**Figure 3 polymers-14-00908-f003:**
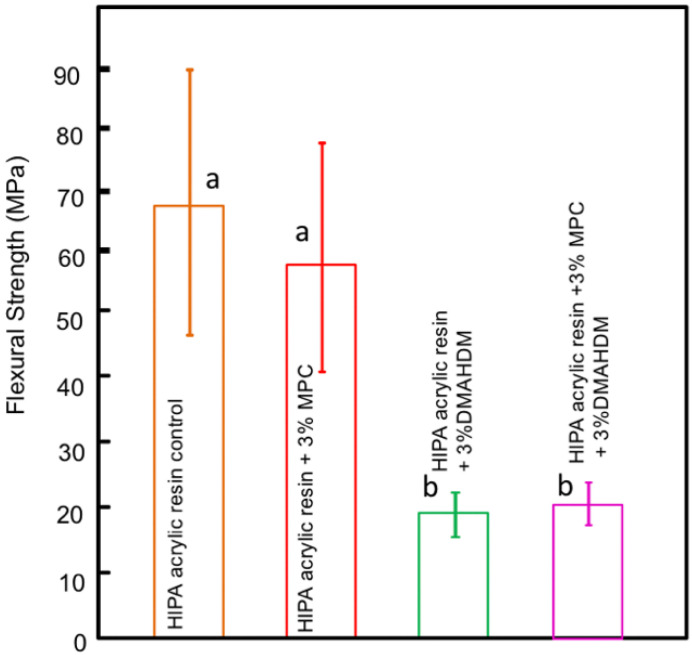
Flexural strength of the acrylic resin; similar letters indicate statistical similarity (Tukey’s test, *p* < 0.05).

**Table 1 polymers-14-00908-t001:** Inclusion and exclusion criteria.

Inclusion Criteria	Exclusion Criteria
Papers written in English	Qualitative and/or quantitative reviews
Papers published from 2010–2021	Case series
Papers studied novel techniques of incorporating anti-fungal and protein repellent agents, both in-vitro or in-vivo	Case reports
	Commentaries
	Letters to the editor
	Interviews
	Updates

## Data Availability

Data is available upon request from the corresponding author.

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
