# Peer review of "How Effective Are Antimicrobial Agents on Preventing the Adhesion of Candida albicans to Denture Base Acrylic Resin Materials? A Systematic Review"

_polymers, 2022, doi:10.3390/polym14050908_

Round 1

Reviewer 1 Report

In the manuscript "How effective are Antimicrobial Agents on Preventing the Adhesion of Candida Albicans to Denture Base Acrylic Resin Materials? A Narrative Review" the authors are presenting the results reported in the last 11 years regarding the efficiency of the denture materials in preventing the adhesion of fungal organisms like C. Albicans.

The paper has potential, but it can be improved.

In material and methods the authors are briefly describing the search conducted and also the inclusion and exclusion criteria. 

To make it easier for the readers, the authors should try to either make a chart or a table in which to have the criteria enumerated.

Also it would be good to show in a chart after each step the number of papers found. 

 Also, it was mentioned that 60 papers were selected, but in the results part are presented less papers.

Although the info are presented on different type of materials, a general conclusion has to be drawn for the data reported.

If the changes mentioned are made, the paper can be considered for publication in the journal.

Reviewer 2 Report

The manuscript is interesting. However, the following are the minor comments.

(i) Why author declares this work as "A Narrative Review".

(ii) Rewrite the abstract in the order of (a) the overall purpose of the study(b) the basic design of the study(c)major findings.

(iii)There is a lack of research gap.

(iv) Make clear objectives.

(v) Reduce the similarity report.

(vi)Rewrite the conclusion by elaborating the exact findings from the results and discussion.

(vii)English correction is very essential throughout the manuscript.

Round 2

Reviewer 1 Report

The authors have made changes to the manuscript. The quality improved and the manuscript can be accepted for publication.